# Nanostructured Hybrid Metal Mesh as Transparent Conducting Electrodes: Selection Criteria Verification in Perovskite Solar Cells

**DOI:** 10.3390/nano11071783

**Published:** 2021-07-09

**Authors:** John Mohanraj, Chetan R. Singh, Tanaji P. Gujar, C. David Heinrich, Mukundan Thelakkat

**Affiliations:** 1Applied Functional Polymers, University of Bayreuth, 95440 Bayreuth, Germany; chetan-raj.singh@uni-bayreuth.de (C.R.S.); tanaji.gujar@uni-bayreuth.de (T.P.G.); David.heinrich@uni-bayreuth.de (C.D.H.); 2Bavarian Polymer Institute (BPI), University of Bayreuth, 95440 Bayreuth, Germany

**Keywords:** metal mesh electrodes, conducting filler layer, ohmic losses, sheet resistance, transparent conducting electrode

## Abstract

Nanostructured metal mesh structures demonstrating excellent conductivity and high transparency are one of the promising transparent conducting electrode (TCE) alternatives for indium tin oxide (ITO). Often, these metal nanostructures are to be employed as hybrids along with a conducting filler layer to collect charge carriers from the network voids and to minimize current and voltage losses. The influence of filler layers on dictating the extent of such ohmic loss is complex. Here, we used a general numerical model to correlate the sheet resistance of the filler, lateral charge transport distance in network voids, metal mesh line width and ohmic losses in optoelectronic devices. To verify this correlation, we prepared gold or copper network electrodes with different line widths and different filler layers, and applied them as TCEs in perovskite solar cells. We show that the photovoltaic parameters scale with the hybrid metal network TCE properties and an Au-network or Cu-network with aluminum-doped zinc oxide (AZO) filler can replace ITO very well, validating our theoretical predictions. Thus, the proposed model could be employed to select an appropriate filler layer for a specific metal mesh electrode geometry and dimensions to overcome the possible ohmic losses in optoelectronic devices.

## 1. Introduction

Recent technological developments transcending the vision of the Internet of Things (IoT) have created a huge global market for “smart” devices such as smart phones, televisions, tablets, watches and many more, which is forecast to grow exponentially over the next few years [1]. Naturally, this progress goes in parallel with the demand for electronic components used in smart devices and in particular, in energy conversion and storage applications. While renewable energy resources are considered for clean and sustainable energy production around the world, solar cells are among the potential technologies that significantly contribute toward the present global energy demands and is anticipated to grow further in the future [2,3]. Both in solar cells and in most of the smart devices containing transparent displays or touch panels, one of the inevitable device stacks is the transparent conducting electrode (TCE). To date, indium tin oxide (ITO) has been ubiquitously employed as TCE in applications requiring high transparency and low sheet resistance. However, with the ever growing demand for TCEs, the scarcity of indium coupled with its cost and energy inefficient production methods and its poor durability in flexible electronic applications prompt the search for appropriate ITO alternatives [4].

Many novel materials such as PEDOT:PSS [5,6], graphene [7], carbon nanotubes [8] and metal nanostructures [9] with low sheet resistance (>1 Ω/□), high transmittance (>80%) in the visible region and high durability have been investigated as TCEs in various optoelectronic applications. Among these, metal nanostructures, in particular, metal mesh electrodes, hold additional promising features like intrinsic high conductivity, transparency in the visible and infrared region, low haze, surface plasmonic effect, high flexibility, and versatile and cost effective preparation strategies, which are best suited for photovoltaic applications [9,10]. For example, Ag nanowires/mesh electrodes are successfully demonstrated as TCEs in semi-transparent organic [11,12,13] and perovskite solar cells [14,15], which in many instances outperform the ITO-based counter parts.

In general, the optoelectronic properties of the metal mesh/network-based TCEs are strictly governed by their geometric and dimensional features such as, (i) metal mesh line width; (ii) pitch size or void area between the subsequent conducting mesh lines and (iii) thickness of the electrode layer [16]. With increasing metal mesh line width, electrical conductivity of the electrode increases and, concomitantly, transmittance decreases; whereas, the trend is reversed upon increasing the pitch size of the network structures. Similarly, increasing the metal electrode thickness reduces the overall sheet resistance, however, it also raises the surface roughness that creates additional shunt paths across the device stacks, impeding the device functionality. Further, the voids within the electrode mesh structure also cause an additional issue in solar cells as the charge carriers reaching these spaces are hardly collected by the electrodes. The lateral transport of charges across the void space towards the conducting mesh lines demands charge carrier diffusion length in tens of micrometer scale, which is not common in many solar cells. In such cases, the accumulated charges in the metal mesh void structures facilitate parasitic charge recombination events that reduce the overall photoconversion efficiency (PCE) of the solar cells [13,16].

To overcome the surface roughness and lateral charge collection issues, the metal mesh structures are often employed in combination with thin conducting filler layers in devices as hybrid electrodes. The filler layer ensures lateral charge collection from the electrode void area to the electrode conducting line; however, this field-driven drift process is accompanied with significant ohmic losses, which is undesirable in any applications, particularly for solar cells. Along these lines, a variety of conducting filler materials such as PEDOT:PSS [15], zinc oxide [13], Al-doped zinc oxide [17] and TiO_2_ [18] have been tested in solar cells in combination with metal mesh electrodes, and a few such combinations have shown promising results.

As of now, the selection of a filler material for a metal mesh electrode is achieved based on the conventional figure of merit determined from the sheet resistance and optical transmission [4]. Based on this, a few studies regarding optimizing the charge collection using metal grids in organic solar cells and their modules are reported [19,20]. Interestingly, in these examples and also during the regular selection of the filler layers, the ohmic losses incurred by the hybrid metal mesh TCEs are not considered and the corresponding influencing parameters are not investigated to the best of our knowledge.

In this work, we use a simple numerical model to determine the optimal hybrid metal mesh geometry for maximizing the current collection in a perovskite solar cell and elucidate its dependency on filler sheet resistance and its effective charge carrier extraction distance, which is a function of the metal mesh electrode pitch size. To verify the theoretical predictions, we fabricated hybrid metal mesh electrodes with well-defined honeycomb-shaped metal network structures with varying pitch size and two different filler materials with different sheet resistance values, and applied them as TCE in perovskite solar cells. The solar cells parameters show a clear reliance of filler sheet resistance on the effective charge carrier extraction distance, which is in line with our numerical model. Based on these results, our model is expected to form the basis for selection rules for TCE, in addition to the existing figure of merit, to fabricate efficient metal mesh electrode/filler combinations, especially for lateral charge collection.

## 2. Materials and Methods

### 2.1. Preparation of Honeycomb Shaped Au and Cu-Network Electrodes on Glass Substrates

Highly periodic, uniform Au-network structures with well-controlled geometric features were prepared on glass substrates using photolithographic technique. On precleaned, ozone treated glass substrates, photoresist LOR5B was spin cast at 5000 rpm for 30 s, followed by annealing at 180 °C for 7 min. Furthermore, photoresist S1813 was spin cast using the same parameters and annealed at 120 °C for 4 min. With the appropriate prepatterned mask on top, the substrates were exposed to ultraviolet (UV) light for 3 s using a mask aligner. Furthermore, the substrates were treated with MF318 solution for 20 s followed by thorough washing in deionized water. The developed substrates were transferred to a physical vapor deposition (PVD) chamber, where 4 nm chromium and 70 nm gold were evaporated successively. After, the substrates were left in a beaker with N-methylpyrrolidone (NMP) solvent overnight, and this was followed by ultrasonication for 4 min to remove all the photoresist. This results in well-defined honeycomb shaped Au-network electrodes on glass substrates. For Cu-network electrodes, the same procedure was followed except approximately 70 nm Cu was evaporated instead of Au in the PVD chamber.

### 2.2. Deposition of Low Temperature TiO_2_ (LT-TiO_2_) and Aluminum-Doped Zinc Oxide (AZO) Filler Layers on Metal Network Electrodes

Compact TiO_2_ layers of approximately 60 nm thickness were prepared on precleaned ITO (reference) and Au-network electrode substrates by spin casting a precursor solution containing 109 μL of titanium tetrabutoxide, 3.4 μL of 37% HCl and 3 μL of ethanol at 2000 rpm for 50 s, followed by annealing at 100 °C for 30 min in ambient conditions. The thickness of the TiO_2_ layers was adjusted to completely cover the metal network electrodes.

Approximately 70 nm thick aluminum-doped zinc oxide (AZO) layers were sputter deposited on ITO, Au-network and Cu-network electrode substrates in a Denton vacuum Explorer^®^ system by employing confocal sputtering technique. A 2 wt% Al-doped ceramic target was used. The substrates were mounted on a stainless-steel holder, which was positioned 60 mm away from the target. The chamber was evacuated to a base pressure of 6 × 10^−6^ bar prior to the sputtering process and during sputtering, the pressure was maintained at 6 × 10^−3^ bar. The sputtering process was carried out with a radio-frequency (RF) power of 137 W for 9 min, and the substrate holder was continuously rotated to ensure the film uniformity.

The thickness of the prepared LT-TiO_2_ and AZO layers deposited on clean dummy glass slides was measured by using a profilometer. The same substrates were used to determine the sheet resistance of LT-TiO_2_ and AZO layers by using four-point probe technique, and the values are >1 G Ω/□ and 7000 Ω/□, respectively. The *R_sh_* of the used ITO reference was determined as approximately 13 Ω/□.

### 2.3. Fabrication of Perovskite Solar Cells

All the starting materials for solar cell preparation were purchased from Sigma–Aldrich (St. Louis, MO, USA) unless otherwise specifically stated, and used as received. The hole transporting material 2,2′,7,7′-tetrakis-(*N*,*N*-di-p-methoxyphenylamine) 9,9′-spirobifluorene (spiro-OMeTAD) was purchased from Merck KGaA, Darmstadt, Germany.

For CH_3_NH_3_PbI_3_ (MAPI) film deposition, 1 M PbI_2_ and methylammonium iodide were dissolved in dimethylformamide and stirred for at least 2 h at room temperature. To facilitate the substrate wetting properties of sputter prepared AZO layer, 60 μL of phenyl-C61-butyricacid (PCBA 0.1 mg/mL) in 1,2-dichlorobenzene solution was spin cast on AZO layer at 2000 rpm for 50 s. Once the MAPI precursor materials were completely dissolved, 80 μL of this solution was spin cast on LT-TiO_2_ and AZO substrates at 3000 rpm for 50 s. While the substrates started spinning, right after 8 s, 200 μL of toluene (orthogonal solvent) was continuously dripped onto the substrates. Furthermore, the substrates were annealed at 100 °C for 25 min on a hot plate inside a N_2_ filled glovebox, resulting in an approximately 400 nm thick uniform MAPI layer. Successively, a hole transporting layer was deposited onto the perovskite layer by spin casting a chlorobenzene solution of spiro-OMeTAD (72.3 mg/mL), 43.2 μL of 4-tert-butyl-pyridine (TBP), and 26.3 μL of LiTFSI (520 mg/mL in acetonitrile) at 2000 rpm for 50 s in the glovebox. These substrates were stored overnight in a dry box (relative humidity < 10%) to facilitate the air doping of spiro-OMeTAD layer. Finally, approximately 70 nm thick gold back contact was deposited in a PVD chamber, completing the solar cells. The active area of the fabricated devices is either 0.09 or 0.16 cm^2^ defined by the cross-section area of the bottom and top electrodes.

Characterization of the solar cells was carried out by using an Oriel solar simulator under AM 1.5, 1000 W m^−2^ conditions and a Keithley 2400 source meter. The current–voltage (I–V) curves were measured from +1.5 V to 0.5 V and back at the scan rate of approximately 130 mV s^−1^ under N_2_ atmosphere. The photovoltaic parameters such as *J_sc_*, *V_oc_*, fill factor (FF) and PCE were determined from the corresponding I–V curves using a home-built software.

## 3. Results

One of the major advantages of metal mesh electrodes is their excellent transparency (>85%), which is a result of their geometry of thin conducting mesh lines with void spaces in between. With increasing distance between the successive metal mesh lines (pitch size), the transparency of the electrode structure improves linearly, making it appealing for transparent and semi-transparent optoelectronic applications. However, in devices that demand charge collection or injection into the void area such as solar cells or organic light emitting diodes (OLEDs), a trade-off between the transparency and conductivity is achieved by controlling the pitch size, and an addition of a conducting filler layer ensures the maximum lateral charge collection or injection, naturally, at the expense of voltage. The impact of such current and voltage losses on organic solar cells performance is addressed in a few reports. For example, Galagan et al. addressed the influence of electrode pitch size in hexagonally structured Ag nanowire grids [19], whereas Jacobs et al. studied different possibilities for incorporating metal wires for improved charge collection in thin film solar cells [21]. Cravino et al., studied the current contribution from neighboring regions around the active area of an organic solar cell by considering the distances between the electrodes and sheet resistance of the charge collecting layer such as PEDOT:PSS used in these devices [22]. The consequences of designing charge collecting metal grids and associated voltage or current loss in modules have also been addressed [20,23]. In spite of these studies, a more realistic picture on ohmic losses in solar cells employing the hybrid metal mesh electrodes can be obtained by combining the sheet resistance (*R_sh_*) of the filler layer, overall charge carrier density in the device and the distance over which the charge carriers needs to be transported (*l*) from the void area to the conducting metal mesh line. To address the voltage and current losses originating from the hybrid metal mesh TCE, we adopted a numerical expression similar to Cravino et al. [22], which they used to estimate additional charge collection from neighboring regions in organic solar cells, and the expression is,
(1)$$\Delta V=\int^{\;}\frac{R_{sh}\;.\;J_{sc}\;.\;l\;.\;w_{el}}{w_{filler}}\;dl\;$$
where Δ*V* is voltage loss, *R_sh_* is sheet resistance of the filler layer, *J_sc_* is photocurrent density, *l* is the distance over which the charge carrier needs to be transported through the filler layer (charge extraction length ≈ about half of the pitch size), *w_el_* is the width of the metal mesh line and *w_filler_* is the effective width of the filler layer depending on the geometry as well as *w_el_* value.

The solution for this expression can be derived using a model system. Consider a perovskite solar cell employed with a honeycomb metal mesh hybrid electrode showing the maximum *J_sc_* of 20 mA/cm^2^. Here, the value of *w_filler_* is assumed to be 20*w_el_*, since the pitch size is approx. 20 times the *w_el_* value. Solving Equation (1) for the process of transporting a charge carrier located in the void at “*l*” distance from the conducting mesh lines (as depicted in Figure 1 inset) results in the plot of *l* vs. *R_sh_* as shown in Figure 1 for the voltage loss of maximum 100 mV and minimum 10 mV.

The plot clearly shows the limitations of the possible combinations of *l* and *R_sh_* for the anticipated voltage loss. For example, to extract a charge carrier from a distance of 100 µm at the expense of 10 mV, the filler should have the maximum *R_sh_* of 1 × 10^5^ Ω/□ (blue dotted arrow). However, in the same device for the same *l* value, if one can afford a 10-fold higher voltage loss (100 mV) then the filler with the maximum *R_sh_* of 1 × 10^6^ Ω/□ (red dotted line) can be used to extract the charges. It is also clear from Figure 1 that for the same filler layer (same *R_sh_)*, with increasing *l*, i.e., upon increasing the pitch size of the metal mesh nanostructures, the loss in voltage also increases. In other words, for each threefold increase in distance between the charge carrier and the conducting mesh line, a minimum of 10-fold higher voltage is required to extract this carrier to contribute to the device photocurrent. This is a significant loss for any high-performing optoelectronic devices, in particular, for perovskite solar cells.

On the other hand, reducing Equation (1) to obtain the relation between *R_sh_* vs. *l* for the above mentioned model system results in:(2)$$\frac{1}{R_{sh}}=l^{2}\frac{J_{sc}}{20\Delta V}$$
which clearly evidences an inverse quadratic relationship between the filler *R_sh_* and *l* value. This means, with an order of increase in *l* value, i.e., for a metal mesh structure with an order of increased pitch size requires a filler layer with at least two orders of magnitude reduced *R_sh_* to collect a charge carrier with a same voltage loss value. In absolute terms, for the given conditions, a filler with a sheet resistance of 100 Ω/□ can effectively transport a charge carrier from a distance of approximately 4 mm with 10 mV loss in the voltage. If *l* exceeds beyond 4 mm, to extract the charge carriers from a metal network void without losing further voltage, it is essential to use a filler with further reduced *R_sh_*. These findings are of paramount importance for solar cells as they assist in mitigating the additional parasitic ohmic losses by suggesting the maximum limit of *R_sh_* of the filler layer for a metal mesh structure with a specific pitch size. Interestingly, the plot also helps to predict the performance loss in any optoelectronic device based on the metal mesh electrode geometry and filler layer combination, prior to device fabrication. 

To verify these theoretical findings, we prepared honeycomb-shaped Au-network electrodes on glass substrates with increasing pitch size and *w_el_* using the photolithographic technique, in combination with two different filler layers showing significantly different *R_sh_* values. Figure 2a–c show the lateral scanning electron microscopic (SEM) images of the prepared Au-network electrode structures along with their corresponding *w_el_* and pitch size. The maximum *l* value for the corresponding void spaces is equivalent to the distance between the center of the honeycomb structure and the middle of a metal mesh line, which can be obtained by determining the radius (r = a × √3/2) of inscribed circles in the hexagons, where “a” is the length of a side of the hexagon. The calculated *l* values corresponding to the honeycomb network pitch size are also listed in Figure 2a–c.

It is evident from the images that the prepared honeycomb Au-network structures are uniform and highly periodic with well-controlled geometry and dimensions, as a result of the photolithographic technique. Notably, the geometry fill factor of these metal mesh electrodes was maintained as 10% despite increasing the pitch size, by adjusting the *w_el_* accordingly (5, 10 and 15 μm). This ensures the same sheet resistance value for all the metal mesh structures irrespective of the pitch size. The influence of increasing *l* value as a consequence of network pitch size and *R_sh_* of the filler layers on ohmic losses is evaluated by employing these metal nanostructures as hybrid TCEs in planar perovskite solar cells.

Using these nanostructured electrodes, two sets of planar perovskite solar cells were prepared by employing low temperature TiO_2_ (LT-TiO_2_, *R_sh_* > 1 G Ω/□) and Al-doped ZnO (AZO, *R_sh_* = 7000 Ω/□) as electron transport as well as conducting filler layers. An approximately 60 nm thick LT-TiO_2_ layer was deposited on Au-network electrode on glass substrates by using sol-gel method, which is known to cover the electrode structure uniformly [13]. On the other hand, approximately 70 nm thick AZO layer was prepared on Au-networks via the RF sputtering technique. The conformal coverage of Au-networks by AZO layer is clearly evidenced by the comparative cross-sectional SEM images of a glass substrate containing the Au-network structure before and after AZO deposition as shown in Figure 2d,e.

The selection of LT-TiO_2_ and AZO for this investigation was based on the similarity in their work function (≈4 eV) [24,25], which circumvents possible differences in contact resistance at the Au-network electrode/electron transport layer interface. Also, as shown in Figure 2f, the transmittance of both LT-TiO_2_ and AZO deposited Au network structures is quite comparable (80%) in the visible region, which safely neglects the possible light transmission differences as a reason for corresponding solar cells performance differences. This transmittance value is less than that of the conventional ITO TCE (85%) in the visible region [13]. Further, the stark contrast in their measured sheet resistance values can lead to unambiguous conclusions regarding the correlation of *R_sh_* and *l* of hybrid TCEs. For this reason, both the electrodes and filler layers are designed and prepared in a careful way that the possible differences in the solar cells performance can be attributed solely to the change in the *l* value and *R_sh_* of the conducting filler layers.

Furthermore, planar perovskite solar cells are fabricated on these conducting filler layers by successively depositing photoactive MAPI, hole transporting spiro-OMeTAD layer and gold top electrode (see Materials and Methods). The final device geometry is Au-network electrode/LT-TiO_2_ or AZO/MAPI/Spiro-OMeTAD/Au and is schematically shown in Figure 3a. In addition to the Au-network electrode-based solar cells, individual reference devices were also fabricated with ITO electrodes for LT-TiO_2_ and AZO filler layers under similar experimental conditions.

Prior characterizing the solar cells, the maximum possible *l* values, i.e., effective charge extraction length at the voltage loss of 10 mV for both LT-TiO_2_ and AZO are deduced from Figure 1 based on their *R_sh_* values, and are schematically shown in Figure 3b. The plot indicates that the RF sputtered AZO (*R_sh_* ≈ 7000 Ω/□) can theoretically transport a charge carrier up to approximately 400 μm distance at the expense of 10 mV; whereas, due to very high *R_sh_* value (>1 G Ω/□) of the prepared LT-TiO_2_, the corresponding maximum lateral charge transport distance is determined to be approximately 1 µm for the same voltage. The substantial difference in their effective charge extraction distance is expected to play a vital role in dictating the corresponding solar cell performance. Based on the solar cells results obtained using Au-network/AZO electrode, we also tested the viability of preparing a similar mesh using less expensive Cu instead of Au, applied it in combination with AZO and compared with the devices employing Au-network/AZO hybrid electrode.

The solar cells were mechanically sealed in test chambers under nitrogen atmosphere and characterized for their photovoltaic features. The measured current–voltage (I–V) curves for solar cells prepared on Au-network electrodes with varying pitch size in combination with LT-TiO_2_ and AZO filler layers and corresponding ITO reference devices are exhibited in Figure 4. The determined parameters from the respective I–V measurements are collected in Table 1.

It should be noted that the prepared perovskite device stacks are not optimized for the maximum PCE. Nevertheless, they are employed in these studies as testbeds due to the excellent photovoltaic properties of MAPI, known for high and similar photocarrier density in all the prepared devices, which is one of the essential criteria to derive the relationship shown in Figure 1 and Figure 3. On the other hand, the ease of preparation and high reproducibility of the corresponding photovoltaic parameters of the planar perovskite solar cells enable investigating the impact of intrinsic changes in one of the device stacks on device performance [26]. Therefore, the main discussion is restricted to the relative differences in the photovoltaic parameters of the devices with increasing Au-network pitch size and different filler layers.

## 4. Discussion

Figure 4a,b display the measured I-V curves of the perovskite solar cells prepared on Au-network hybrid TCEs with LT-TiO_2_ and AZO filler layers, respectively. Considering the Au-network/LT-TiO_2_ based perovskite solar cells, they exhibit photovoltaic character, however, the determined *J_sc_*, *V*_oc_ and fill factor (FF) values are remarkably poor, irrespective of the Au-network pitch size (Table 1). On the other hand, with increasing Au-network pitch size from 97 to 293 μm, the *R_s_* value of the respective perovskite solar cells linearly increases from 1500 to 4500 Ω·cm^2^ and *R_shunt_* also increases relatively. This emphasizes the influence of Au-network pitch size on device performance upon employed with a poorly conducting filler layer such as LT-TiO_2_. The reference solar cells (Figure 4c) prepared with LT-TiO_2_ on ITO electrode (ITO/LT-TiO_2_) also show *R_s_* and *R_shunt_* values comparable to a device with the Au-network/LT-TiO_2_ TCE (97 μm pitch size, Table 1), however, it displays *J_sc_* of 11.7 mA/cm^2^, *V_oc_* of 0.99 V and FF of 56% resulting in an overall PCE of 6.5%, which is comparable with the solar cells containing LT-TiO_2_ prepared using a similar method [27]. This clearly suggests that the device stacks in Au-network/LT-TiO_2_ TCE-based solar cells are not limiting the performance. On the other hand, the calculated maximum allowable *l* value for LT-TiO_2_ (approximately 1 µm) based on its *R_sh_* is exceeded well in all the prepared Au-network electrode structures under investigation. This is anticipated to result in poor lateral charge extraction by LT-TiO_2_ towards the metal mesh lines, and consequently leads to charge carriers’ accumulation and recombination at the electrode/LT-TiO_2_ interface and poor *J_sc_*, *V_oc_* and PCE values. Furthermore, upon comparing the photovoltaic parameters of the devices within the same series, both *J_sc_* and *V_oc_* considerably increase with decreasing *l* value from 141 to 44.3 µm in the hybrid TCEs resulting in 2–3 fold higher PCE values. These experimental observations are in line with the predictions based on our numerical model.

In the case of Au-network/AZO-based hybrid electrodes, the calculated effective charge carrier transport distance (400 µm) of the filler layer is well above the *l* values corresponding to the Au-network structures used. Thus, theoretically, AZO is expected to effectively extract all the photogenerated charge carriers from the Au-network electrode void space without significant ohmic losses. In fact, all the perovskite solar cells prepared on Au-network/AZO hybrid TCEs display comparable photovoltaic parameters including *R_s_* and *R_shunt_* values with those of the ITO/AZO based reference device (Figure 4b,c, and Table 1). With increasing *l* from 44.3 μm (pitch = 97 µm) up to 141 μm (pitch = 293 µm) in Au-network/AZO-based devices, *J_sc_* is maintained at approximately 15 mA/cm^2^ with a slight fluctuation in *V_oc_*. Interestingly, unlike the devices prepared on Au-network/LT-TiO_2_ TCEs, the *R_s_* and *R_shunt_* values of the respective solar cells remain comparable irrespective of the Au-network pitch size (Table 1), suggesting the influence of the conducting filler layer. The fact that there is no current collection loss up to a pitch size of 293 µm is in full agreement with our model, which allows the charge extraction of up to 400 µm for a 10 mV drop. We still observe a slight decrease in *V_oc_* and FF as the pitch size is increased to 293 µm, which could be due to: (i) shunt loss originating from device stack preparation steps, and/or (ii) the effective charge carrier extraction distance of AZO is overestimated as the plots in Figure 1 are derived by using the upper limit of *J_sc_*, *w_el_* and *w_filler_* values, and/or (iii) occurrence of charge carrier recombination events at the perovskite/electron transport layer interface, which generally reduces the *V_oc_* and FF [28].

Using this same strategy, we tested if Earth-abundant and less expensive copper grids in combination with AZO can be used as a hybrid TCE in perovskite solar cells. Figure 4c showing the measured I–V curves of perovskite solar cells containing ITO/LT-TiO_2_ and ITO/AZO reference electrodes, and Au-network/LT-TiO_2_, Au-network/AZO and Cu-network/AZO hybrid TCEs with the pitch size of 196 μm and the determined corresponding photovoltaic parameters listed in Table 1 indicate similar photovoltaic behavior of hybrid Cu-network/AZO based solar cells without any loss in the current collection. This first result is promising and opens the possibility of exploiting the combination of copper grid electrodes and a variety of appropriate filler layers as hybrid TCEs for semi-transparent perovskite solar cell applications.

## 5. Conclusions

In this work, we emphasized that charge carrier extraction process from the voids of metal mesh electrode using conducting filler layer depends heavily on the sheet resistance of the latter and metal electrode pitch size. The consequent losses in current and voltage can be considered as ohmic losses, which could be detrimental to optoelectronic device performance, if the sheet resistance of the filler layer is very high. To calculate the extent of such losses, a simple numerical expression involving the distance over which the carrier needs to be transported (*l*), sheet resistance of the filler layer (*R_sh_*) and charge carrier density of the device is proposed. By solving this expression, the interplay between *R_sh_* and *l* of the filler layer for conditions relevant for solar cells applications is derived. The corresponding plots suggest the following: (i) with increasing *l* value, voltage loss also increases for the same filler layer; (ii) with increasing *R_sh_* of the filler layer, the effective *l* decreases quadratically for an assumed voltage loss value. These results are experimentally verified by fabricating two sets of TCEs based on Au-network electrodes with varying pitch size and with LT-TiO_2_ and AZO filler layers, and testing them in perovskite solar cells. Evidently, Au-network/LT-TiO_2_-based TCE exhibited extremely poor charge extraction behavior compared to the respective reference device, and displayed a clear trend between increasing *l* values and *J_sc_* and *V_oc_* parameters, which are in excellent agreement with the predicted model. On the other hand, using an improved conductor as filler layer, an extremely good charge collection behavior was observed in devices using Au-network/AZO as well as Cu-network/AZO based TCEs. These observations highlight the correlation and impact of *R_sh_* of the filler layer on lateral charge extraction efficiency towards the metal mesh electrodes, and validate the predictions from our numerical model. These experimental results suggest that the proposed model can help in determining the ohmic losses in optoelectronic devices originating from the hybrid metal mesh TCEs prior to device preparation. In addition, the derived relation between the effective charge carrier extraction distance and the filler layer sheet resistance could enable selecting the appropriate conducting filler layers for specific metal mesh electrode geometry and dimensions aimed for a variety of transparent and semi-transparent optoelectronic applications.

## Figures and Tables

**Figure 1 nanomaterials-11-01783-f001:**
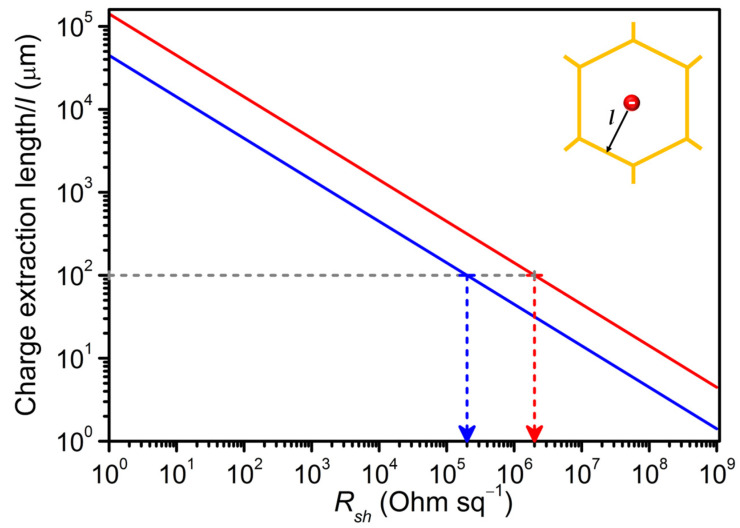
Relation between the sheet resistance (*R_sh_*) of the filler layer and charge carrier extraction length (*l*) from a void space in the metal network electrode as shown in the inset scheme. The plots are derived by assuming *J_sc_* of 20 mA/cm^2^ and Δ*V* of 100 mV (red) or 10 mV (blue) in a perovskite solar cell. The horizontal grey line intercepting red and blue lines notifies the required *R_sh_* value of the filler to collect the charges from 100 μm distance with the voltage loss of 100 and 10 mV, respectively.

**Figure 2 nanomaterials-11-01783-f002:**
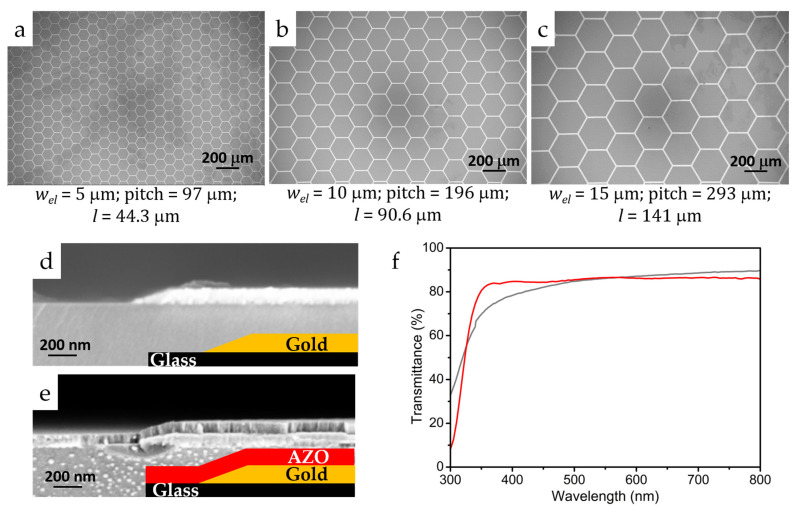
Scanning electron microscopy (SEM) images of Au-network electrodes with the pitch size of (**a**) 97 μm, (**b**) 196 μm and (**c**) 293 μm deposited on glass substrates showing uniform honeycomb structures with well-defined geometry and dimensions over a large range; cross-sectional SEM images of (**d**) Au-network structure on a glass and (**e**) conformal covering of Au-network structure on glass by sputter-deposited aluminum-doped zinc oxide (AZO) layer and corresponding schemes as inset; (**f**) transmittance spectra of LT-TiO_2_ (grey) and AZO (red)conducting filler layers deposited Au-network (pitch = 97 μm) electrodes on glass substrates.

**Figure 3 nanomaterials-11-01783-f003:**
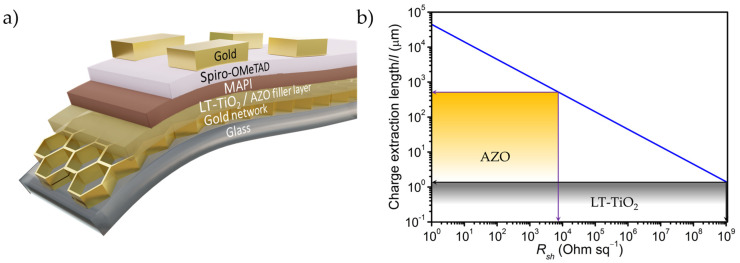
(**a**) Schematic representation of the planar perovskite solar cell device configuration and corresponding stacks used in this work; (**b**) Effective charge extraction length of LT-TiO_2_ and AZO filler layers corresponding to their *R_sh_* values deduced for 10 mV voltage loss from Figure 1.

**Figure 4 nanomaterials-11-01783-f004:**
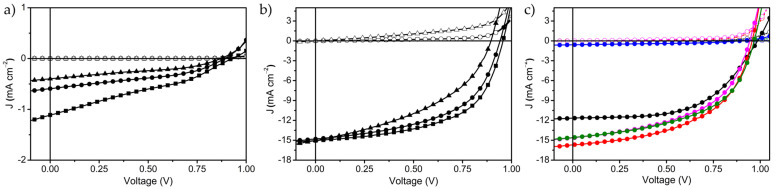
Dark (open symbol-lines) and light (filled symbol-lines) I–V curves of planar perovskite solar cells prepared on Au-network electrodes with the pitch size of 97 μm (square-line), 196 μm (circle-line) and 293 μm (triangle-line) containing (**a**) LT-TiO_2_ and (**b**) AZO filler layers; (**c**) comparison of dark (open symbol-lines) and light (filled symbol-lines) current–voltage (I–V) curves of perovskite solar cells prepared on reference ITO/LT-TiO_2_ (black-circle-line), ITO/AZO (red-circle-line) and hybrid Au-network/LT-TiO_2_ (blue-circle-line), Au-network/AZO (magenta-circle-line) and Cu-network/AZO (green-circle-line) electrodes with 196 μm pitch size.

**Table 1 nanomaterials-11-01783-t001:** Photovoltaic parameters of the planar perovskite solar cells prepared on Au-network with increasing charge carrier extraction length, and LT-TiO_2_ and AZO conducting filler layers.

Electrode and Filler Layer	Pitch Size (μm)	*l* (μm)	J_sc_ (mA/cm^2^)	V_oc_ (V)	FF (%)	PCE (%)	*R_s_* (Ω·cm^2^)	*R_shunt_* (Ω·cm^2^)
ITO/LT-TiO_2_ (Reference)	-	-	11.7	0.99	56	6.5	1380	1028
Au network/ LT-TiO_2_	97	44.3	1.11	0.93	31	0.32	1562	938
	196	90.6	0.59	0.89	39	0.21	2500	2215
	293	141	0.40	0.87	37	0.13	4545	2839
ITO/AZO (Reference)	-	-	15.7	0.96	52	7.9	448	395
Au network/ AZO	97	44.3	15.1	0.96	56	8.2	136	473
	196	90.6	14.8	0.94	53	7.4	117	377
	293	141	15	0.90	43	5.8	42	183
Cu network/ AZO	196	90.6	14.6	0.97	52	7.4	151	371

## Data Availability

The data presented in this study are available on request from the corresponding author.

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
