# Peer review of "Nanostructured Hybrid Metal Mesh as Transparent Conducting Electrodes: Selection Criteria Verification in Perovskite Solar Cells"

_nanomaterials, 2021, doi:10.3390/nano11071783_

Round 1
Reviewer 1 Report
Recommendation: Major revision
In this work, Mohanraj et al report that charge carrier extraction process from the voids of metal mesh electrode using conducting filler layer heavily depends on the sheet resistance and pitch size. They systematically verified their model with perovskite solar cell on their metal mesh electrode. A series of characterization techniques have been used to investigate the achievement. Although the sophisticated methods used in this work, some issues need to be addressed before this paper is published.
- Authors provided a image of conformal covering of Au-network structure on glass by sputter deposited AZO layer in Figure 2e. But authors should provide a cross-section images of AZO (and LT-TiO2) on their metal mesh. Otherwise, we do not know whether filler materials are properly filled into the metal mesh.
- According to Table 1, smaller pitch size shows better solar cell performance. How’s the efficiency of solar cells when the pitch size is smaller than 97 um for Au network/AZO and 196 um for Cu network/AZO? I also want to know why authors used 196 nm pitch size for Cu network/AZO device. I think authors didn’t fully optimize their device.
- Authors need to provide the hysteresis for the solar cells with tested electrodes. Then, they can discuss more about the performance of their electrode.
- Authors should provide the stability of the solar cells with tested electrodes. Sometimes, electrodes boost a degradation of the perovskite solar cells. This information will be helpful for all readers of this article.
Reviewer 2 Report
The author systematically found the optimized metal network design with metal oxide layer and compared the performance of photovoltaic cells depending on the dimension of metal network, which is very helpful for developing flexible opto-electronic devices. However, following issues should be ironed out before its publication
1) The author mentioned that transmittance of metal network based TCE plays important role to determine the performance of device with it. However, the author do not include any data about transmittance of demonstrated TCE depending on the its dimension. Please provide the transmittance of TCE.
2) As author mentioned in page 2, the height of metal network should be optimized to prevent solar cell from shunt losses. However, the author only investigated case with 70 nm thick metal network. It there any reason for choosing 70 nm as a thickness of metal network?
Moreover, is there any leakage loss in your cells? Although the author provided SEM image of TCE in the Figure 2 d and e, the analysis of possible leakage path for device, including perovskite morphology is not properly provided in the manuscript.
3) In the figure 4 and table 1, could you add series and shunt resistance of perovskite solar cell, which is affected by sheet resistance of TCE ? If these are provided, it would be better to understand the relationship between sheet resistance of TCE and device performance.
4) Why is the FF of perovskite solar cell with 293 um pitch gold network and AZO layer lower than others? In the discussion section, the author insisted that it is attributed to low Voc due to poor sheet resistance AZO and recombination loss in the interface of perovskite/electron transporting layer. However, its I-V curve shows strong shunt loss rather than recombination losses. Please clarify the origin of decreased efficiency of the perovskite solar cell.
Reviewer 3 Report
The manuscript: “Nanostructured Hybrid metal mesh as Transparent Conducting
Electrodes: Selection criteria verification in perovskite solar cells " by J. Mohanraj et al. reports the results of selection criteria verification of nanostructured hybrid metal mesh as transparent conducting tlectrodes alternatives for indium tin oxide in perovskite solar cells. The authors have analyzed a conducting filler layer to collect charge carriers from the network to minimize current and voltage losses. The authors used a general numerical model to correlate the sheet resistance of the filler, lateral charge transport distance in network voids, metal mesh line width and ohmic losses in optoelectronic devices. To verify this correlation, the authors prepared gold or copper network electrodes with different line widths and different filler layers and applied them as transparent conducting electrode in perovskite solar cells. It was shown that the photovoltaic parameters scale with the hybrid metal network TCE properties and a Au-network or Cu-network with AZO filler can very well replace ITO. The proposed model could be employed to select the appropriate filler layer for specific metal mesh electrode geometry and dimensions to overcome the possible Ohmic losses originating from the metal mesh hybrid electrode geometry.
The authors have carried out a rather interesting study that is of interest to researchers working in the field of perovskite solar cells. The main discussion is related to the relative differences in photovoltaic parameters of devices with increasing Au-grid step size and different filler layers. The importance of this approach is obvious.
At the same time, there is a commentary on the manuscript.
First, the main problem is related to the photovoltaic parameters of the devices. The authors noted that the prepared cells of perovskite devices are not optimized for SCs, which are 4-5 times lower than the record values ​​for perovskite SCs. The authors used these structures as tests because of the excellent photovoltaic properties of MAPI. However, I am not sure if this is the correct approach. The PCE of the test samples must be much higher in order to check the Au mesh spacing and the different filler layers. Can the same test be done with high PCE SC samples? Second, what about the I-V hysteresis and environmental sustainability of the investigated perovskite SC with and without a hybrid metal mesh as transparent conductor electrodes? There are no other comments.
In conclusion I suggest that the subject of this manuscript suits the topic of Nanomaterials, however the manuscript needs minor optional revision. The manuscript can be considered for publication in Nanomaterials.
Round 2
Reviewer 1 Report
-